

# Identifying biomarkers for evaluating wound extent and age in the contused muscle of rats using microarray analysis: a pilot study

Na Li[1], Chun Li[1], Dan Li[1], Li-hong Dang[1], Kang Ren[1], Qiu-xiang Du[1], Jie Cao[1], Qian-qian Jin[1], Ying-yuan Wang[1], Ru-feng Bai[2] and Jun-hong Sun[1]

[1] School of Forensic Medicine, Shanxi Medical University, Jinzhong, China
[2] Key Laboratory of Evidence Science, China University of Political Science and Law, Beijing, China

## ABSTRACT

Wound age estimation is still one of the most important and significant challenges in forensic practice. The extent of wound damage greatly affects the accuracy and reliability of wound age estimation, so it is important to find effective biomarkers to help diagnose wound degree and wound age. In the present study, the gene expression profiles of both mild and severe injuries in 33 rats were assayed at 0, 1, 3, 24, 48, and 168 hours using the Affymetrix microarray system to provide biomarkers for the evaluation of wound age and the extent of the wound. After obtaining thousands of differentially expressed genes, a principal component analysis, the least absolute shrinkage and selection operator, and a time-series analysis were used to select the most predictive prognostic genes. Finally, 15 genes were screened for evaluating the extent of wound damage, and the top 60 genes were also screened for wound age estimation in mild and severe injury. Selected indicators showed good diagnostic performance for identifying the extent of the wound and wound age in a Fisher discriminant analysis. A function analysis showed that the candidate genes were mainly related to cell proliferation and the inflammatory response, primarily IL-17 and the Hematopoietic cell lineage signalling pathway. The results revealed that these genes play an essential role in wound-healing and yield helpful and valuable potential biomarkers for further targeted studies.

## INTRODUCTION

In violent cases, an accurate estimation of wound age, which describes the time interval between trauma infliction and death, is one of the most challenging points in forensic pathology (*Casse et al., 2016*; *Trautz et al., 2019*; *Wang et al., 2016*). The precise estimation of wound age contributes to rebuilding the scene of the case, demarcating the scope of suspects, determining the nature of the cases, and finally facilitating the investigation process (*Du et al., 2020*; *Sun et al., 2017*). However, the determination of wound age is influenced by many additional factors such as age, gender, concomitant diseases, therapy, and the extent of the wound damage making it even harder to ensure the accuracy and

Corresponding authors
Ru-feng Bai, bairufeng@cupl.edu.cn
Jun-hong Sun,
junhong.sun@sxmu.edu.cn

reliability of wound age prediction (*Grellner & Madea, 2007*; *Li et al., 2018*). Among these factors, the extent of a wound, which is highly variable among the muscles examined in forensic practice, may be the most important.

It is well known that wound age estimation mainly depends on the time-dependent expression patterns of the molecules involved in damage repair. Although skeletal muscle holds an innate capacity for self-repair after injury, fibrosis, or fatty muscle deposits with significant functional deficits, often occur in the case of severe injury. Fibrosis is related to both the different molecular responses after injury as well as the severity of the wound. Thus, the temporal expression pattern of molecules responding to injury varies with both the stage of the healing process and the severity of the injury. Most studies have concentrated solely on the timing of morphological and biochemical changes during wound healing (*Hassan Gaballah et al., 2016*; *Ishida et al., 2015*; *Wang et al., 2015*) and have ignored the effect of the level of wound damage. A careful evaluation of the extent of a wound is necessary to effectively determine wound age.

With the rapid development of life science and technology, particularly high-throughput technology, the measurement of damage parameters has improved, and more and more potential biomarkers for estimating the time of injury have been studied (*Casse et al., 2016*; *Li et al., 2018*). Microarray technology, which allows the simultaneous evaluation of the expression profiles of thousands of genes, has increasingly contributed to our understanding of many biological processes (*Komamura et al., 2003*; *Li et al., 2017*) and also shows great potential for identifying valuable biomarkers of the time-dependent expression for wound age estimation. The main problem with using microarray technology is that the number of genes greatly exceeds the number of samples following a microarray experiment, leading to the main concern of how to identify genes of interest from such a large gene set.

Fortunately, the least absolute shrinkage and selection operator (LASSO) regression model was introduced with the potential to solve this problem. In a LASSO-penalized regression, as log λ (a tuning parameter) changes, the corresponding coefficients of certain genes are reduced to zero, indicating that their effects on the model can be omitted because they are shrunk parameters (*Tibshirani, 1997*). LASSO regressions allow fewer hub genes to be obtained for further analysis. Moreover, a time-series analysis, based on the multivariate empirical Bayes' statistic (MBstatistic), is sufficient to find genes with large differences in their temporal patterns between conditions and then rank them (*Tai & Speed, 2009*). We applied the LASSO method in our study to evaluate the most informative prognostic mRNA biomarkers according to their relative contribution to the prognosis of damage degree; we also used a time-series analysis to identify genes with large differences in temporal profiles for wound aging.

In a previous study, we established the muscle contusion models of rats caused by blunt force attack at different heights, and evaluated the gross and histological morphological changes over time as well as verified the repeatability of the model (*Bai et al., 2017*). Different numbers of inflammatory cells and different sizes of hemorrhage and edema were observed in mild and severe contusion muscles. In this study, we aimed to explore the molecular changes at the transcriptional level of mildly and severely contused muscle and identify valuable potential biomarkers for evaluating the extent of the wound as well as

wound age. Because basic physiological responses are similar in both humans and animals, the predictive genes for wounds found in this study will provide valuable information for further forensic practice.

## MATERIALS & METHODS

### Animal model and tissue preparation

All procedures were performed according to the ''Guiding Principles in the Use and Care of Animals'' (NIH Publication No. 85-23, Revised 1996) and were approved by the Institutional Animal Care and Use Committee of Shanxi Medical University of China (Batch number of rats: SCXK (Jin) (2009-0001)). Animals received humane care in conformity to the principles in the Guide for the Care and Use of Laboratory Animals protocol, published by the Ministry of the People's Republic of China (issued on June 4, 2004). The laboratory personnel were given permission to conduct this study by the Committee after they attended and completed training on how to use experimental animals ethically. All male Sprague Dawley rats were purchased from the Animal Center of Shanxi Medical University. This study was carried out in compliance with the ARRIVE guidelines and evaluated and approved by the Institutional Animal Care and Use Committee of Shanxi Medical University of China with approval number 2019sll002.

An animal model was established based on previous studies (*Bai et al., 2017*). In total, 33 male Sprague Dawley rats (aged 10–12 weeks), weighing 250–300 g, were placed in a cage with rat chow and water, under a 12-hour light-dark cycle in a temperature-controlled room (22–24 °C) with a relative humidity of 40–60%. No adverse events were observed. All rats were randomly divided into three groups:

(1) Control group ($n = 3$): without any treatment before sampling.

(2) Mild contusion groups ($n = 15$): a 500 g counterpoise fell freely from a height of 15 cm down a clear Lucite guide tube onto the thigh muscles of the right posterior limb with the potential energy of the contusion totaling about 1.46 J/cm$^2$ (the formula: Ep = mgh).

(3) Severe contusion groups ($n = 15$): a 500 g counterpoise fell freely from a height of 50 cm, with the potential energy of contusion at about 2.58 J/cm$^2$ (the formula: Ep = mgh).

The animals in the mild and severely contused groups were further divided into five subgroups (1-, 3-, 24-, 48-, and 168-hour post-injury; n = 3/subgroup) for microarray analysis. Before muscle contusion, the rats were anesthetized with pentobarbital sodium to minimize the suffering of the rats during the experimental period. The rats in the contusion groups were sacrificed at the above timepoints with a lethal dose of pentobarbital (350 mg/kg body weight, intraperitoneal injection). Approximately 100 mg of muscle was sampled from the wound site and equally divided into two parts from each rat. For the control rats, specimens were harvested from the same site after anesthetization with a lethal dose of pentobarbital. All the muscle samples from 33 rats were frozen immediately with liquid nitrogen for the microarray analysis.

A further 56 rats were randomly allocated to a control group ($n = 8$) and 4-, 8-, 12-, 16-, 20-, and 24-h (n = 8/group) mild contusion groups for qPCR. Before contused, the

right posterior limbs of the rats with pentobarbital were shaved by a depilatory agent (Nair; Carter Wallace, New York, NY, USA) and placed on an experimental table in a supine position. After damage, the rats were transferred into clean cages with food and water.

## RNA extraction

Total RNA was extracted as described in previous study (*Du et al., 2020*). Specifically, we isolated the RNA from the skeletal muscle specimens (approximately 50 mg each) using RNAiso Plus 9108 (Takara Bio, Shiga, Japan), according to the manufacturer's protocol. The quantity (ng/mL) and purity of RNA were accessed by a microplate reader (Infinite M200 Pro; TECAN, Zurich, Switzerland), and the integrity of the RNA was further analyzed with an Agilent 2100 instrument (Agilent Technologies, Palo Alto, CA, USA) using an Agilent RNA 6000 Nano kit. Only RNAs with OD260/OD280 ratios ranging from 1.8 to 2.2 and a RIN > 7.0 were selected for the microarray and the RT-PCR (reverse transcription polymerase chain reaction).

## Microarray analysis

Total RNA was used to synthesize sense-strand cDNA (sscDNA) using the Ambion WT Expression Kit. Then, fragmentated and labeled sscDNA samples (using the Affymetrix® GeneChip® WT Terminal Labeling Kit) were hybridized to Gene 1.1 ST Array Plates. Washed and stained arrays were scanned using the GeneChip Scanner 3000 7G.

## Data processing

Statistical significance of gene expression differences between the contusion and control groups was analyzed using one-way ANOVA in the Affymetrix Transcriptome Analysis Console software version 4.0.1. The expressions of mRNA with fold change > 2 in mean expression compared with the control group and a *P*-value of < 0.05 and FDR (false discovery rate) < 0.05 were considered differentially expressed genes (DEGs).

A heatmap and an unsupervised hierarchical clustering analysis were performed to cluster DEGs in the mild and severe groups, respectively, using the online software Morpheus (https://software.broadinstitute.org/morpheus). Principal component analysis (PCA) was employed to evaluate the importance of DEGs, distinguish the extent of contusions, and visualize the effect of the DEGs we selected in wound age estimation based on different wound degrees using SIMC A-P (ver. 14.1; Umetrics, Malmö, Sweden). T-distributed Stochastic Neighbor Embedding (t-SNE) was performed by R package "tsne" (version 0.15).

A least absolute shrinkage and selection operator (LASSO) regression model was then used to select key genes through a penalized likelihood approach (*Tibshirani, 1997*). In the model, as log λ (a tuning parameter) changed, the regression coefficients penalize the size of the parameters, so that the coefficient estimates of unimportant variables can shrink towards zero (*Yan et al., 2020*). Moreover, the "time course" R package was applied to find genes of interest based on the multivariate empirical Bayes model. It was reported that this model was superior to the traditional F-statistic for the analysis of time-series data (*Tai & Speed, 2009*).
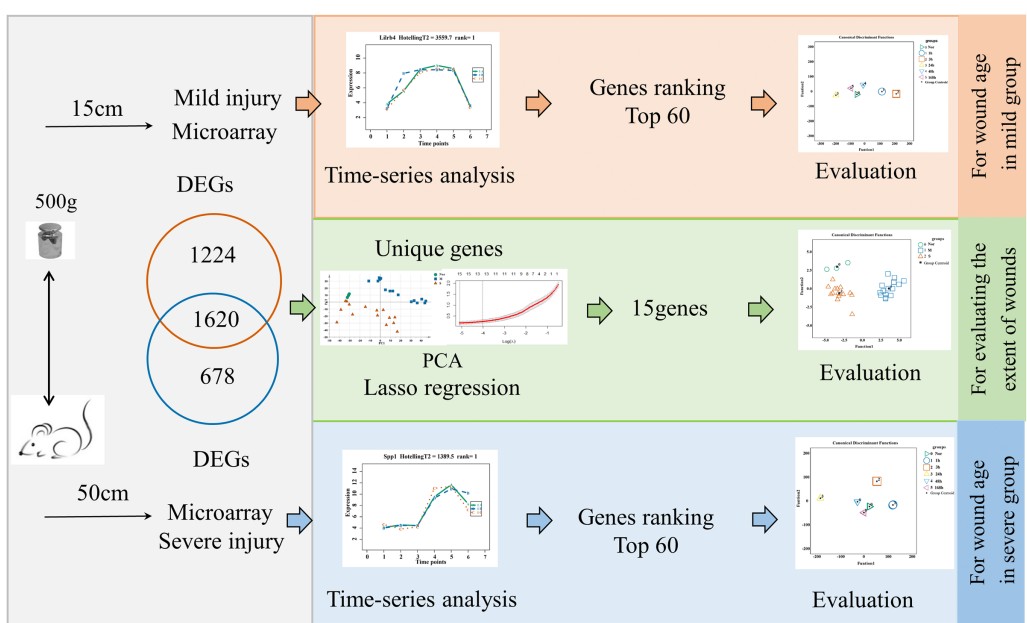

**Figure 1** Flow diagram of the experimental design and data analysis used in the present study.

Gene ontology (GO) annotation was conducted using the Gene Ontology Consortium (http://geneontology.org) and the Kyoto encyclopedia of genes and genomes (KEGG) pathway annotation was performed by clueGo plugin (https://apps.cytoscape.org/apps/cluego) (*Bindea et al., 2009*). The protein-protein interaction (PPI) network was built by STRING 11.0 (https://string-db.org) and the networks were visualized and exported from Cytoscape 3.7.2 (*Shannon et al., 2003*). In addition, the top 10 hub genes in the PPI network were selected according to the degree method using the plug-in CytoHubba in Cytoscape (*Chin et al., 2014*). A Fisher discriminant analysis (FDA) was performed using SPSS software (SPSS version 24.0).

## Quantitative real-time PCR

We prepared the reaction mixtures using the Premix Ex Taq$^{TM}$ kit (Takara Biotechnology Co., Ltd., Dalian, China) and then put them into the Bio-Rad CFX384 fluorescence Quantitative PCR system (CFX384; Bio-Rad, Hercules, CA, USA) . Amplification was performed using a 2-step cycling program according to the following conditions:40 cycles of 5 s at 95 °C, 40 s at 58–60 °C. Two reference (RPL13, ribosomal protein L13; RPL32, ribosomal protein L32) and two target genes were simultaneously assayed, as they were in previous studies (*Dang et al., 2020*; *Du et al., 2020*). Each reaction mixture consisted of Taq DNA polymerase (12.5 μL), upstream and downstream primers and probes (0.65 μL; eight primers and four probes), 10% (v/v) DMSO (2.0μL), deionized water (1.2 μL), and cDNA (1.5 μL). The primers and probes that were used are listed in Table S1.

The flow diagrams, shown in Fig. 1, summarize the experimental and data analysis procedures used in the present study.

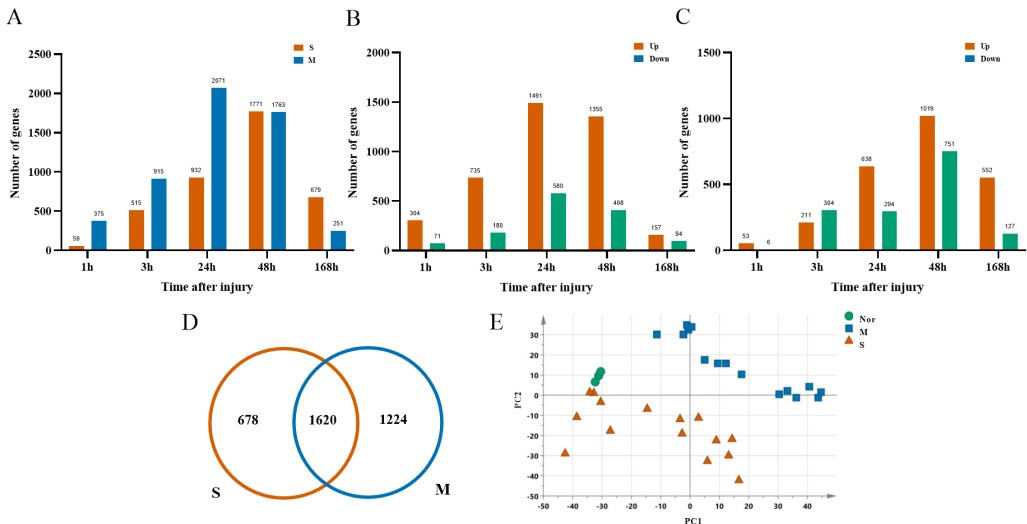

**Figure 2   Dynamic transcriptome changes in mildly and severely contused muscle.** (A) The number of DEGs with $P < 0.05$ and 2-fold change at each time point in the mild and severe contusion groups. (B) The number of upregulated and downregulated DEGs in the mild injury groups. (C) The number of upregulated and downregulated DEGs in the severe injury groups. (D) Venn diagram showing that there were 1,620 common genes shared and 1,902 unique genes between the mild and severe contusion groups. (E) The PCA result of unique genes between the mild and severe contusion groups; samples were colored according to the degree of damage. "Nor" means normal groups (in bluish green); "M" means mild injury groups (in blue); "S" means severe injury groups (in vermilion).

# RESULTS

## Transcriptional profiles in mildly and severely contused muscle

After ruling out probes with low expressions, 16,005 probes were selected from 33 rats for the next analysis. There were 2,844 DGEs presented in the mild contusion groups and 2,298 DEGs in the severe contusion groups compared with the control group (Figs. 2A, 2D). The number of DGEs found in mild injury was larger than that found in severe injury except at 168 h. And at each time point, compared with the control group, the number of upregulated DGEs was more than down-regulated even with different degrees of injury (Figs. 2B, 2C). A PCA analysis of unique genes showed that the mildly and severely contused groups were clearly separated from each other, indicating that the severity of damage contributes the most to the variance in the data set (Fig. 2E).

## Hub genes to distinguish the extent of the injury

The unique DEGs expressed between the mild and severe damage groups were then used to assess the extent of wound damage. In a principal component analysis of unique genes, over 70% of the variance among the DEGs was retained in the first five principal components (PCs) (Fig. 3A). The score of each principal component is displayed in Figs. 3B–3C and Figs. S1A–S1C. Although the variance of the first principal component (PC1) was the largest of the first five principal components, PC2 captured most of the variance related to the genes expressed between the mild and severe groups (Figs. 3A–3C). As in Fig. 3C, the
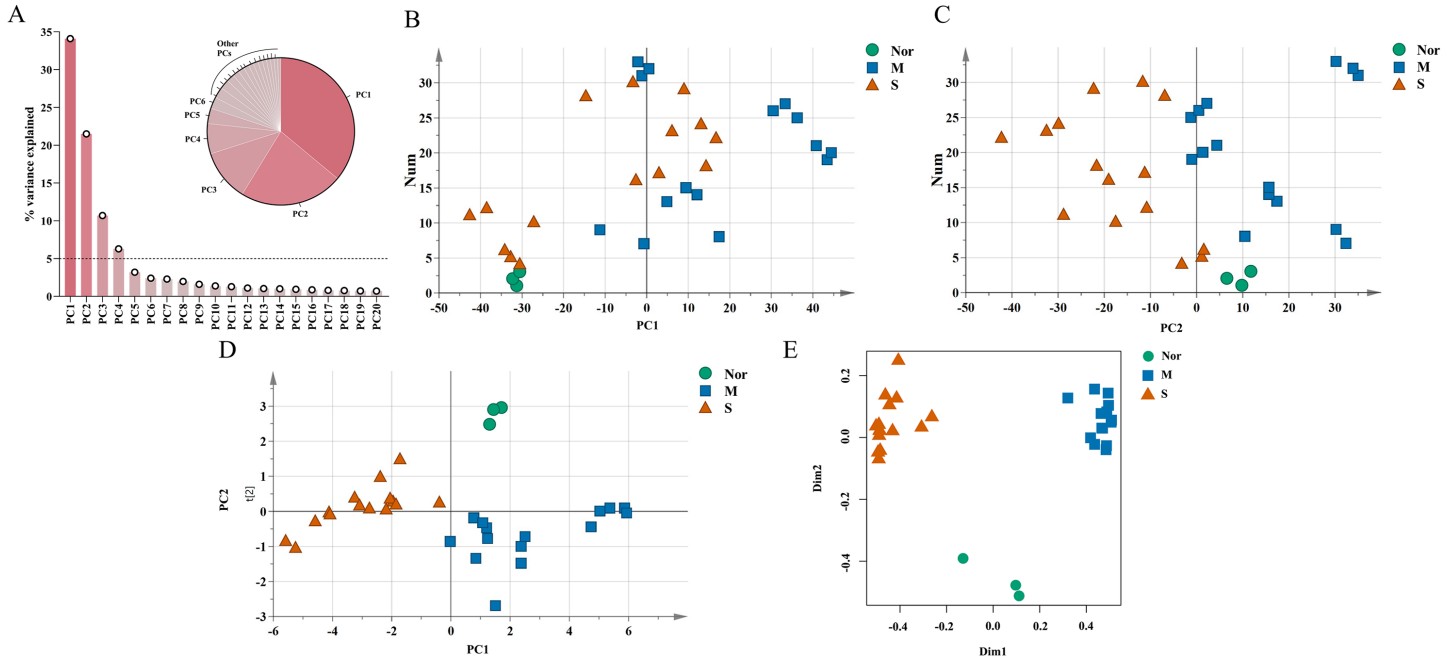

**Figure 3  Identification of genes that drive the differences in gene expression between mild and severe injury.** (A) Percentage of variance explained by each principal component. (B–C) The score plots of principal component (PC)1 and PC2. (D–E) PCA (D) and Random Forest (E) of the 15 prognostic DEGs showed that the control group and the mild and severe contusion groups were clearly separated from each other. "Nor" means normal groups (in bluish green); "M" means mild injury groups (in blue); "S" means severe injury groups (in vermilion).

mild and severe contusion groups were distributed in different areas on PC2, so the top 200 leading genes in PC2 were selected for further analysis.

A least absolute shrinkage and selection operator (LASSO) regression model was then used to select key genes (*Tibshirani, 1997*). We conducted the LASSO algorithm with the R "glmnet" package (*Friedman, Hastie & Tibshirani, 2010*), and the optimal tuning parameter (λ) was chosen to achieve the minimal partial likelihood deviance in the cross-validation plot (Figs. S1D–S1E). The 15 genes with non-zero corresponding coefficients were entered into the multivariate model (Table S2). The PCA score plots showed that the control group and the mild and severe contusion groups were clearly separated from each other, indicating that the expression of 15 genes differed in mildly and severely contused muscle (Fig. 3D). This conclusion was also supported by a random forest analysis (Fig. 3E).

### The screening of candidate markers for injury time estimation in mild contusion
#### *Candidate markers with time-series analysis*
An unsupervised hierarchical clustering analysis of 2,844 DEGs in the mild contusion groups showed that the expression profiles of mild injury at the six observed time points could be segregated into three distinct clusters: the first cluster included contused muscle 24 and 48 h after injury; the second cluster included contused muscle 0, 1, and 168 h post-injury; and the third cluster included contused muscle 3 h after injury (Fig. 4A). PCA of all DEGs in mild injury also showed that the samples displayed a temporal separation

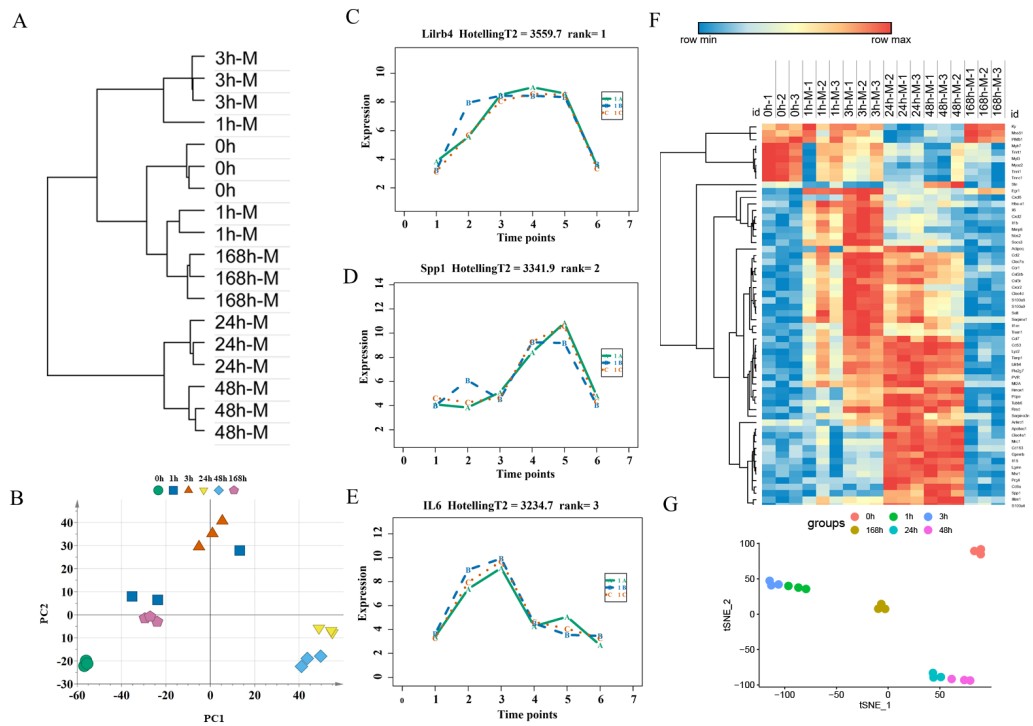

**Figure 4** Time-series analysis of differentially expressed genes in mild contusion groups. (A) Unsupervised hierarchical clustering of gene expression profiles. (B) Principal component analysis of differentially expressed genes in all biological replicates. (C–E) Top three significant genes according to HotellingT2 value; 1A (in bluish green), 2B (in blue), and 3C (in vermilion) represent the three biological replicates at each time point. (F) Heatmap of the expression levels of the top 60 differentially expressed genes in the mild injury groups, showing that the expression can be clustered into four temporal patterns. (G) The performance of t-SNE based on these 60 genes.

in the PCA space (Fig. 4B). One sample at 1 h that was clustered with the 3 h samples may be related to the fact that 1 h and 3 h are very close and hierarchical clustering and PCA are unsupervised analyses with limited discriminating ability. In addition, the fact that the samples of 1 h were grouped with 168 h may be due to the similarity between the beginning and the end of the inflammation stage as both had a few DEGs with similar expression profiles. Generally, the transcriptome profiles exhibited time-dependent patterns that allowed us to identify useful markers of wound age.

For wound aging, the "time course" package in R language was used to obtain genes with large differences in their temporal patterns over time after injury based on the empirical Bayes model. Finally, the top 60 genes were chosen as candidate genes by their HotellingT2 value in descending order (Table S3). The top 3 genes with different expression profiles in mild injury were visualized and displayed in Figs. 4C–4E. A heatmap for the top 60 DEGs revealed that these genes were expressed in four different temporal patterns, and the gene expression in these four clusters were significantly changed over the period after injury (Fig. 4F). Moreover, a t-SNE analysis of these 60 DEGs showed different distributions for

different contusion groups and that they had the predictive potential for wound estimation in mild contusion groups (Fig. 4G).

*Bioinformatics analysis of candidate markers*

To associate the changes of gene expressions with biological functions in mild injury, we conducted a GO enrichment analysis of the top 60 genes to determine the enriched GO terms. DEGs were categorized by their function into biological processes, cellular component, and molecular function. $P < 0.05$ was used as a threshold to select significant GO categories in the Gene Ontology Consortium and for KEGG pathway analysis in clueGo plugin (Figs. 5A, 5B). In mild injury groups, we found that the top 3 categories of biological processes were related to interspecies interaction between organisms, inflammatory response, and response to an external stimulus. GO categories of cellular components were associated with myofibril, contractile fiber, and sarcomere, and those related to molecular functions were relevant to cytokine activity, signaling receptor activator activity, and receptor–ligand activity, suggesting the importance of these genes during mild injury estimation. To identify the functional relationship among genes, a PPI network of 60 genes was created and all genes had a STRING score of 0.7 or greater. We used cytoHubba in Cytoscape to determine the hub genes, all of which play critical regulatory roles in the PPI network. The top 15 genes with a high degree of connectivity were: Il1b, Il6, Ccl2, Il18, Cxcl2, Timp1, Serpine1, Hmox1, Cxcr2, Cxcl6, Mmp8, Ccl7, Spp1, Ccr1 and Tnni1, which were on the inside of the circle (Fig. 5C).

## The screening of candidate markers for severe injury time estimation
*Candidate markers with time-series analysis*

For expression profiling of the severely contused group, we used the same methods as in the mildly contused groups. The PCA and unsupervised hierarchical clustering analysis of 2298 DEGs in the severe contusion groups revealed that samples from each time point were clustered into a single group. Cluster analysis indicated two distinct transcriptome patterns of mild injury in a time-dependent manner, which was supported by PCA results (Figs. 6A, 6B). The empirical Bayes model was also conducted using the 'time course' R package to rank genes based on their expression over time. Lastly, the top 60 genes were acquired by their HotellingT2 values in descending order for wound age estimation in severe contusion (Table S4). The top 3 genes with different expression profiles in severe injury were visualized and displayed in Figs. 6C–6E. A heatmap for the top 60 DEGs revealed that the majority of genes were highly expressed in the late stage of injury (Fig. 6F). Additionally, a t-SNE analysis of these 60 DEGs showed a separation among the contused groups, revealing that their expression differed over time and deserved to be explored further for wound age estimation (Fig. 6G).

*Bioinformatics analysis of candidate markers*

In severely contused muscle, the top 3 categories of biological processes were related to defense response, interspecies interaction between organisms, and response to external biotic stimulus. Those of cellular components were associated with the external side of the plasma membrane extracellular space, side of membrane extracellular space, side of

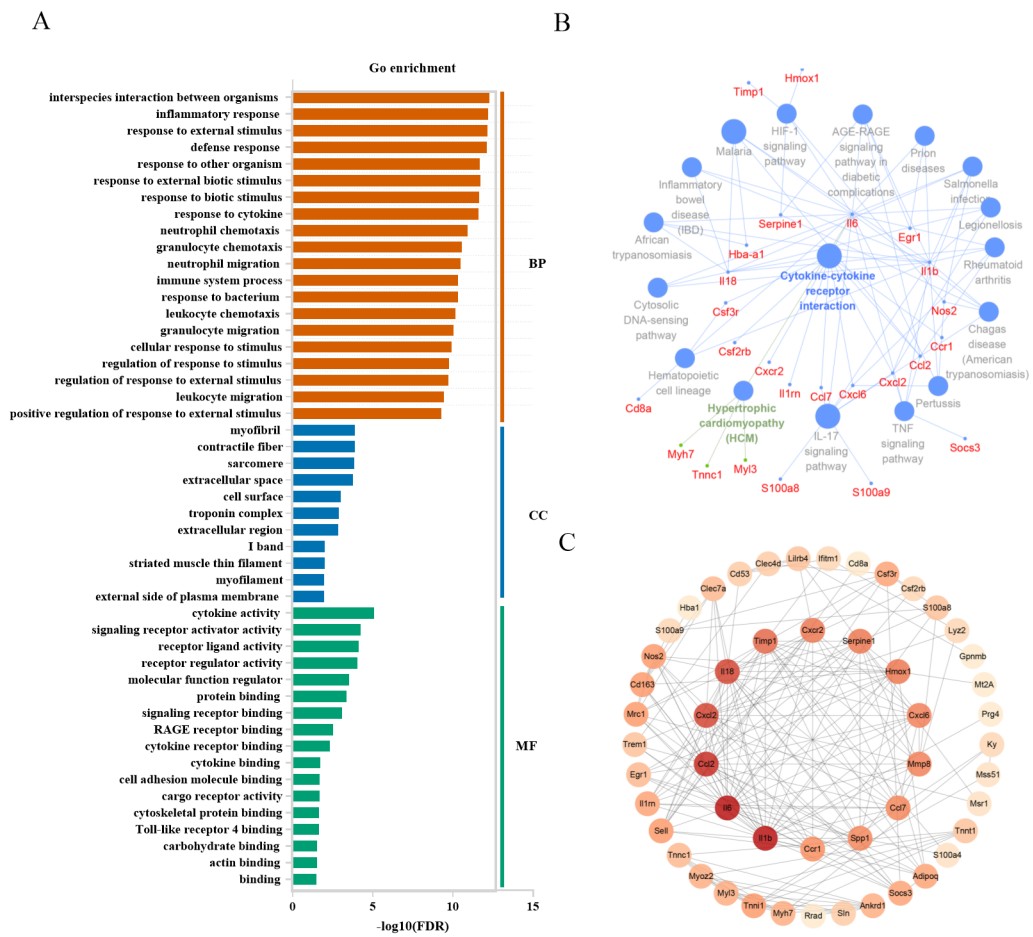

**Figure 5** **Functional analysis of top 60 differentially expressed genes in mild injury.** (A) Go enrichment; BP, biological progress; CC, cellular component; MF, molecular function. (B) KEGG analysis using the clueGo plugin in Cytoscape 3.7.2. (C) Functional network of genes. The more reddish the node, the higher the connectivity degree of the node.

membrane, and those of molecule functions were relevant to signaling receptor binding, cytokine activity, and signaling receptor activator activity (Fig. 7A). A KEGG analysis of the top 60 genes in the clueGo plugin revealed that there were common pathways in both mild and severe injury estimation such as the IL-17 signaling pathway and Hematopoietic cell lineage as well as unique pathways in mild injury estimation including cytokine-cytokine receptor interaction and the TNF signaling pathway (Fig. 7B). Most of these pathways were related to the process of injury and repair, in which genes regulated the process of inflammation, for example, Cxc16, Ccr1, Csf3r, Cxcr2, Il18, Cc17, Cc12, Il6, and Cxc12. Moreover, some pathways associated with bacterial infection appeared such as Malaria and Salmonella infection, which may have close associations with the inflammatory response. Using the STRING platform, the PPI network was built of these 60 genes in severe injury with a STRING score ≥ 0.7. The top 15 hub genes determined by cytoHubba were: Cd68,

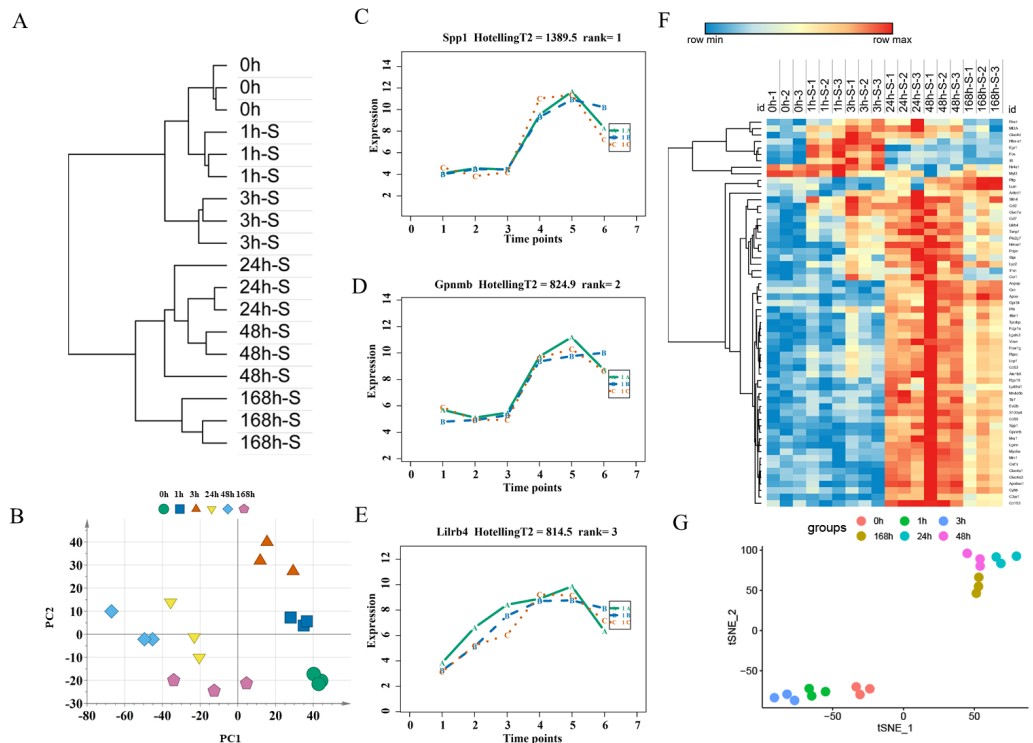

**Figure 6** Time-series analysis of differentially expressed genes in severe contusion groups. (A) Unsupervised hierarchical clustering of gene expression profiles. (B) Principal component analysis of differentially expressed genes in all biological replicates. (C–E) Top three significant genes according to HotellingT2 value; 1A (in bluish green), 2B (in blue), and 3C (in vermilion) represent the three biological replicates at each time point. (F) Heatmap of the top 60 differentially expressed genes in the severe injury groups. (G) The t-SNE analysis of these 60 genes in the severe injury group.

Il6, Ptprc, Tyrobp, Ccl2, Csf1r, C3ar1, Cybb, Fcgr1a, Cd53, Lgals3, Timp1, Spp1, Tlr7, and Cd163 (Fig. 7C).

## Validation of markers in mild injury by quantitative real-time PCR

We validated the microarray data by examining the expression of Mt2a, Rrad, Gpnmb, and Timp1 (shared genes of selected top60 genes of mild and serve injury) in the mild contusion groups using RT-qPCR (Figs. 8E–8H). In previous studies, we verified the time-dependent expression of Mt2a mRNA in the contused skeletal muscle of rats within 36 h by RT-qPCR (*Fan et al., 2017*) (Fig. 8E). *Ling et al. (2016)* found that Mt2a is closely associated with oxidative stress, mediated by subcellular pathways of mitochondria, ER, lysosomal, and lipidosome, as well as the MAPKs (ERK, JNK, and p38) signal. In present study, we continued to validate the expression of Rrad, Gpnmb, and Timp1 (Figs. 8F–8H, Table S5) in 56 rats. The expressions of Mt2a, Gpnmb, and Timp1 detected by RT-qPCR were elevated in a time course manner, which was consistent with the results of the microarray (Figs. 8A–8D). Although the profiles of Rrad were not identical to what we obtained from the microarray, mRNA of Rrad was upregulated within 24 h compared with the control group and corresponded to the Affymetrix analyses. Collectively, the consistency between

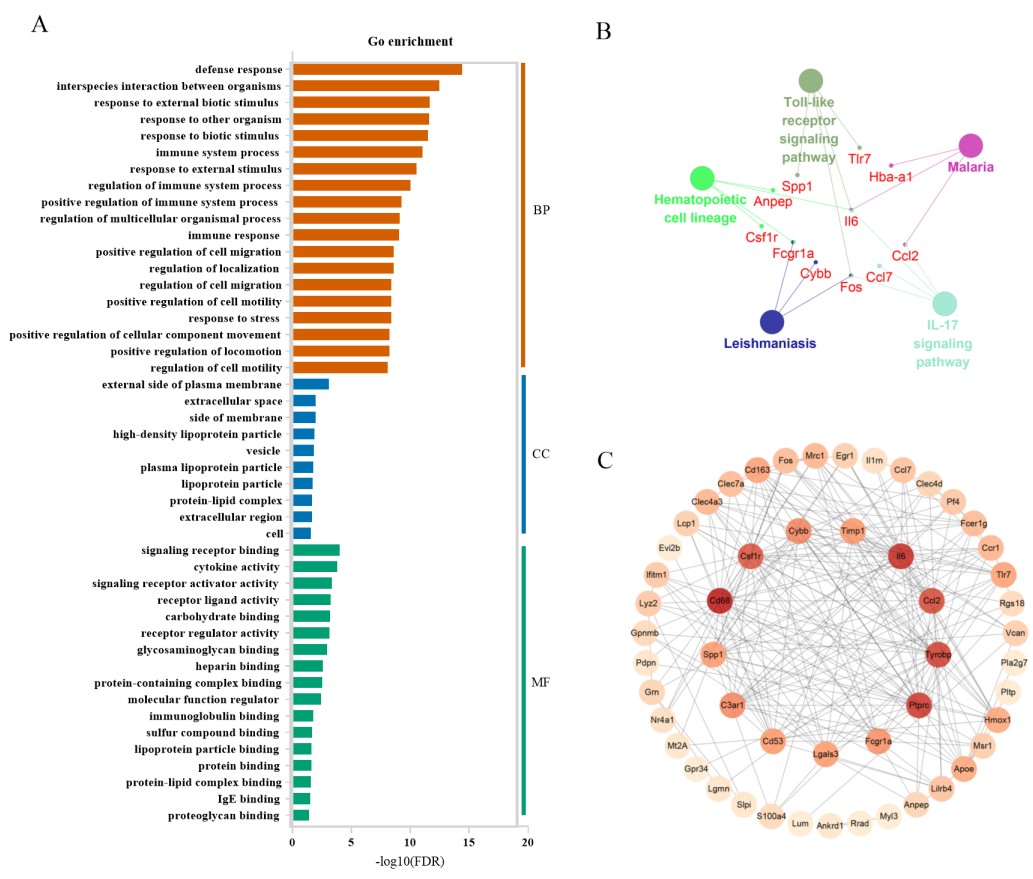

**Figure 7  Functional and pathway enrichment of differentially expressed genes in severe injury groups.**
(A) Go enrichment; BP, biological progress; CC, cellular component; MF, molecular function. (B) KEGG
analysis by clueGo plugin in Cytoscape 3.7.2. (C) Functional network of these genes. The more reddish the
node, the higher the connectivity degree of the node is.

the results of qPCR and the microarray analyses confirmed the reproducibility of the
microarray data.

## Performance of screened genes in Fisher discriminant analysis

To validate the predictive and diagnostic performance of selected indicators, a Fisher
discriminant analysis (FDA) was employed. Based on the expression of 15 hub genes,
the canonical functions can differentiate between the control, mild, and severe groups
(Fig. 9A). The classification results showed 90.9% accuracy for cross-validated grouped
cases (seen in Table S6), indicating that the molecules may respond differently during the
repair process of mild and severe injuries. For wound age estimation in mild injury, Lgmn,
PVR, Hmox1, Egr1, Socs3, Ankrd1, Cd8a, Pfkfb1, Msr1, Tnni1, and Myl3 of the top 60
candidate genes showed good performance with only one wrongly classified sample of 168
h in model of FDA (Fig. 9B). In severe injury, Apoe, Hmox1, Clec4a1, Egr1, Rrad, Il1rn,
Akr1b8, C3ar1, Pltp, Ly49si1, and Lilrb4 of the top 60 genes were included in the FDA
model using the stepwise method and the Wilks' lambda option. The results showed that

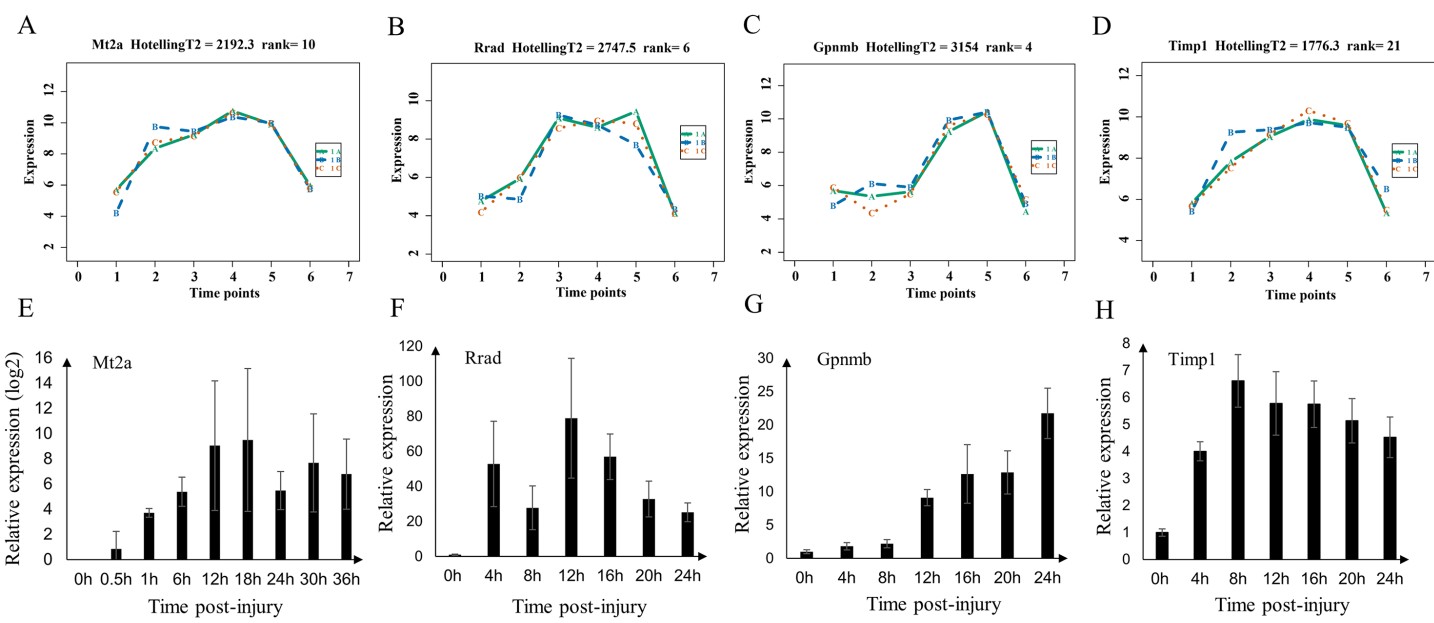

**Figure 8** **Expression level of Mt2a, Rrad, Gpnmb and Timp1 in microarray analysis (A–D) and RT-qPCR (E–H) in mild group.** (A–D) 1A (in bluish green), 2B (in blue), and 3C (in vermilion) represent the three biological replicates in microarray analysis.

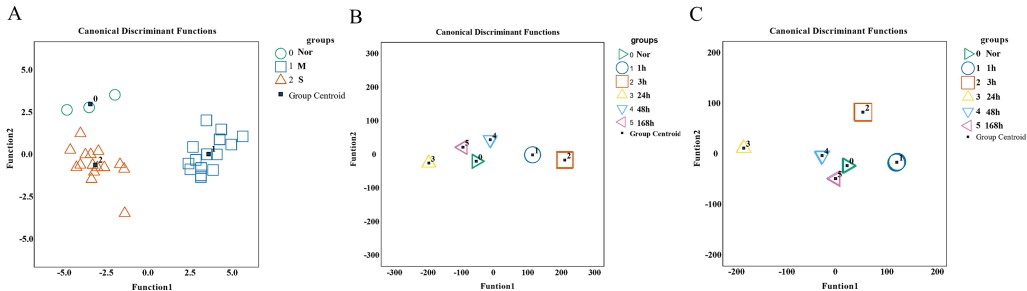

**Figure 9** **Performance of genes we selected for wound degree and age estimation in Fisher discriminant analysis (FDA) model.** (A) The canonical discrimination functions for the degree of damage. 0○ control group, 1□ mild injury group, 2△ severe injury group (B, C) The FDA function for wound age estimation in mild and severe injury groups. The Wilks' lambda option was used in the Stepwise method. (0▷ 0 h, 1○ 1 h, 2□ 3 h, 3△ 24 h, 4▽ 48 h, 5 ◁ 168 h).

all the samples were classified correctly (Fig. 9C, Table S7–S8). Collectively, the 15 hub genes and 120 candidate genes we provided showed great potential for determining the extent of the wound and wound age and are valuable candidates for further study.

## DISCUSSION

Although a number of research projects have investigated how to determine the time of injuries, significant barriers remain in the application of useful biomarkers to establish the chronology of an injury in forensic practice. There are many factors that contribute to

wound age estimation, but one factor that plays as essential role in wound age estimates is the extent of the wound. However, most studies focus on the changes at the morphological and molecular levels in bruises over time without considering the impact of the force used to inflict the trauma. *Barington & Jensen (2016)* made an attempt to compare the changes of bruises inflicted with a low, moderate, and high force, but they only explored bruises within 8 h at gross and histological levels which is still far from practical applications.

In cases of violent injury, the severity of muscle bruising often varies among individuals and among sites of damage within the same individual. Forensic pathologists mostly estimate wound age based on the time-dependent changes of the molecules' response to an injury and the morphological transformation caused by these molecules (*Dang et al., 2020*; *Wang et al., 2016*). Many wound healing-related biomarkers have recently been explored for wound aging (*Hassan Gaballah et al., 2016*; *Ishida et al., 2018*; *Kubo et al., 2014*); however, only a few markers have sufficient diagnostic power for high-accuracy wound age estimation, which is partly because the molecules respond differently at different levels of damage.

Therefore, it is important to identify useful biomarkers to evaluate the extent of a wound and then to determine injury time. In this study, experimental animal models of muscle contusion with different heights of blunt force were developed, and microarray technology was utilized to identify a series of markers over time at the transcriptome level. Our study aimed to screen useful markers to evaluate the extent of a wound, estimate damage time, and to be used for further studies of wound age.

We first identified a total of 3,522 DEGs in mildly and severely contused muscle compared to normal skeletal muscle in rats. According to PCA results, we found that unique genes can better distinguish mild and severe injury than the genes that are shared between the two groups. In order to discover indicators that contribute more to the discrimination of damage degrees, the top 200 leading genes in PC2 of five principal components were chosen for subsequent analysis (Fig. 2C). In order to select the most powerful predictive prognostic genes for identifying the extent of the wound damage, the least absolute shrinkage and selection operator (LASSO) regression model was used for variable shrinkage. Finally, 15 key genes were selected. The PCA score plots showed that the control group and the mild and severe contusion groups were clearly separate from each other, indicating that the expression of the 15 key genes differed in mildly and severely contused muscle (Figs. 3D–3E). This conclusion was also supported by a random forest analysis.

In addition, most of these 15 genes have been previously reported to play a vital role in cell proliferation and the inflammatory response. Reducing APIP expression could help treat systemic inflammatory response syndrome, a whole-body inflammatory state that can occur in response to infection (*Ko et al., 2012*). Fam220a genes have been identified as novel target genes for miR-489-3p and miR-92a-3p associated with renal injury and hypercholesterolemia (*Wiese et al., 2019*). Myadml2, which encodes myeloid-associated differentiation marker-like 2, may influence the ability of dairy cows to resist the inflammation response of mastitis (*Chen et al., 2015*). Trim72, with a RING finger domain, a B-box, two coiled-coil domains, and a SPRY domain, is specifically expressed in

the plasma membrane of skeletal and cardiac muscle cells and is transcriptionally activated by the synergism of MyoD (or myogenin) and MEF2 (*Cai et al., 2009*; *Jung & Ko, 2010*; *Lee et al., 2010*). Ntn1, which encodes netrin 1, is a target of the Wnt/APC oncogenic pathways connected with the cell proliferation and inflammation response of hypoxia (*Rosenberger et al., 2009*; *Zhou et al. 2015*). GPS2, as a molecular guardian, is necessary for precise control of inflammatory responses involved in immunity and homeostasis (*Cardamone et al., 2012*; *Venteclef et al., 2010*). Yeats4, which encodes the YEATS domain containing 4, is closely related to cell proliferation in the cancer process (*Ji, Zhang & Yang, 2017*; *Jixiang et al., 2017*; *Pikor et al., 2013*). Hivep2, which encodes human immunodeficiency virus type I enhancer-binding protein 2, is associated with developmental delay, intellectual disability, and dysmorphic features (*Steinfeld et al., 2016*). These 15 genes have the potential to diagnose the extent of wounds and deserve to be explored further.

Having identified genes for evaluating the severity of a wound, we next used a time-series analysis based on the "time course" package in R language to find indicators for wound aging in the mild and severe contusion groups, respectively. As a result, the top 60 genes with large differences in temporal patterns among different time points post-injury were acquired for each group based on HotellingT2 values (Tables S3–S4). There were 30 genes that were shared by the mild and severe injury groups, and their expression profiles were strongly correlated with time in both contusion groups. This indicates that there were similar molecular and cellular responses during the skeletal muscle repair process, although these responses were affected by the severity of the injury. These shared genes were primarily involved in the inflammatory response and in cytokine activity such as Ccr1, Il6, Ccl7, Pdpn, Spp1, Ccl2 and Timp1, which play vital roles in the wound healing process. The expression of Mt2A, one of the 30 shared genes identified, was verified in our previous study (*Fan et al., 2017*). In this study, the expression patterns of the shared genes (Rrad, Gpnmb, and Timp1) were also examined using qPCR, which were generally similar to microarray data; the results indicated that the data produced *via* the Affymetrix array were reliable.

In the top 60 genes of mild injury, we found that the top 3 categories of biological processes were related to interspecies interaction between organisms, and inflammatory response to an external stimulus. GO categories of cellular components were associated with myofibril, contractile fiber, and sarcomere, and those of molecule functions were relevant to cytokine activity, signaling receptor activator activity, and receptor–ligand activity, suggesting the importance of these genes during mild injury estimation. However, in the top 60 genes of severe injury, the top 3 categories of biological processes were related to defense response, interspecies interaction between organisms, and response to external biotic stimulus. Those of cellular components were associated with the external side of the plasma membrane, and those of molecule functions were relevant to signaling receptor binding, cytokine activity, and signaling receptor activator activity. There were also common pathways in mild and severe injury estimations such as the IL-17 signaling pathway and the Hematopoietic cell lineage, as well as unique pathways in mild injury estimation including cytokine-cytokine receptor interaction and the TNF signaling pathway

(Figs. 5C, 7C). The genes identified in both mild and severe injury were correlated with cell proliferation and the process of inflammation, indicating that time-dependent genes play an important role in repair after injury.

This study identified different criteria and methods of biomarker identification for evaluating the extent of wounds and wound age. In order to identify genes to distinguish the extent of the injury, the unique DEGs, which were only significantly changed in the mild or severe damage groups compared with the control groups, were selected to identify the differences between mild and severe contusions. The expression profiles of the 15 genes selected for evaluating the extent of the damage differed more in mildly and severely contused muscle than they did when evaluating wound age. Conversely, for wound age estimation, our goal was to find genes expressed over time that contained different information at different time points in mild or severe injury. Therefore, the top 60 genes with a large difference in their temporal patterns were screened for wound age estimation in the mild and severe contusion groups, respectively. All these efforts were made to help improve diagnostic accuracy and narrow the margin of error for wound age estimation based on distinguishing the degree of damage.

In this study, microarray analyses were performed to characterize the transcriptional expression changes of mildly and severely contused muscle collected at different time points postinjury. After obtaining thousands of DEGs, PCA, LASSO regression, and time-series analyses were used to select the most powerful predictive genes. Finally, 15 genes for evaluating the extent of wound damage and the top 60 genes for wound age estimation in mild and severe injury were respectively screened. A functional enrichment analysis revealed that these selected genes are actively involved in the muscle repair process and deserve to be addressed and explored further in relation to wound age estimation.

## CONCLUSIONS

For an accurate assessment of wound age, while considering the effect of wound severity, we carried out a transcriptome-based microarray assay to obtain the expression profiles of wound healing genes following mild and severe muscle contusions. Using statistical and bioinformatics analyses, diagnostic genes were identified for evaluating the extent of wounds and wound age in future studies. However, there are limitations to this study. The sample size for the microarray analysis was small; further research with larger samples is necessary to validate more biomarkers in contused muscle with different degrees of damage.

### Funding

This project was supported by grants from the National Natural Science Foundation of China (Grant Numbers: 81971795 and 81373241). The funders had no role in study design, data collection and analysis, decision to publish, or preparation of the manuscript.

## Grant Disclosures

The following grant information was disclosed by the authors:
The National Natural Science Foundation of China: 81971795, 81373241.

## Competing Interests

The authors declare there are no competing interests.

## Author Contributions

- Na Li, Chun Li, Dan Li and Li-hong Dang performed the experiments, analyzed the data, prepared figures and/or tables, and approved the final draft.
- Kang Ren analyzed the data, authored or reviewed drafts of the paper, and approved the final draft.
- Qiu-xiang Du, Jie Cao, Qian-qian Jin, Ying-yuan Wang, Ru-feng Bai and Jun-hong Sun conceived and designed the experiments, authored or reviewed drafts of the paper, and approved the final draft.

## Animal Ethics

The following information was supplied relating to ethical approvals (i.e., approving body and any reference numbers):

All procedures were performed according to the "Guiding Principles in the Use and Care of Animals" (NIH Publication No. 85-23, Revised 1996) and were approved by the Institutional Animal Care and Use Committee of Shanxi Medical University of China [Batch number of rats: SCXK (Jin) (2009-0001)].

## Ethics

The following information was supplied relating to ethical approvals (i.e., approving body and any reference numbers):

The Institutional Animal Care and Use Committee of Shanxi Medical University of China approval to carry out the study within itsfacilities (Ethical Application Ref: 2019-sll-002).

## Data Availability

The microarray data are available in GEO: GSE162565.

## Supplemental Information

Supplemental information for this article can be found online at http://dx.doi.org/10.7717/peerj.12709#supplemental-information.

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
