# Peer review of "Identifying biomarkers for evaluating wound extent and age in the contused muscle of rats using microarray analysis: a pilot study"

_PeerJ, doi:10.7717/peerj.12709_

## Round 0.1 · original submission · Major Revisions

It would be particularly important to address the concerns and suggestions of the first reviewer.

·

Basic reporting

There are several typo errors, it is recommended that a native English speaker review the manuscript.
The supplemental tables information does not coincide with the text (i.e. page 13, line 276).
It is confusing in the title of the manuscript about the evaluation of the extent of the wounds since the authors only classify the wounds in mild and severe depending on energy applied to the rat´s muscle, and they don´t mention other characteristics of this type of wounds in their model. I think the authors should clarify this point.
I suggest the authors include an English translated version of the Institutional Animal Care and Use Committee.
Figures 2b-i, could be as supplemental figures.
It is not mentioned to what correspond the line colors in figures 3c-e and figures 5c-e. Also, the name of the genes should be mentioned in these figures.
Better legend of figures is required.

Experimental design

The authors should explain how the identification of hub genes distinguish the extent of the injury, and how they are related with the markers found for injury time in mild and severe contusion.
It is not clear the criteria used to select the Rrad, Gpnmb, and Timp1 for RT-qPCR validation. Also, the authors should validate some of the fifteen genes found differentially expressed in supplemental table 1.
They should mention in the text the housekeeping genes and the genes explored in the validation with qRT-PCR. Also, it is missing the primers and probes in Table S1.

Validity of the findings

I think the authors should discuss about the grouping of 1h-M with 3h-M, and 1h-M with 168h-M in figures 3a and 3b.
It would be interesting if authors mention which genes of the 60 most significant found in mild and severe groups are shared between them.

Additional comments

No comments.

Reviewer 2 ·

Basic reporting

.

Experimental design

.

Validity of the findings

.

Annotated reviews are not available for download in order to protect the identity of reviewers who chose to remain anonymous.

·

Basic reporting

The authors discussed a very important and interesting topic and they included all results relevant to their hypothesis, however, there are some comments on certain parts of the manuscript that need to be modified

Experimental design

The experimental design, although clear, but will be better to show it in a simplified figure

Validity of the findings

the research topic shows novelty. all data have been provided. conclusions are well-stated and linked to the original research question

Additional comments

English editing of the whole manuscript is required
figure 1 the resolution needs to be improved

---

## Round 0.2 · accepted · Accept

Thanks for the revised manuscript. The revisions have been considered sufficient for the paper to be accepted for further processing.

·

Basic reporting

no comments

Experimental design

very nice experimental design

Validity of the findings

no comments

Additional comments

The authors have taken the reviewer's comments into consideration and made all required changes to their manuscript.